# Appraisal of Intraoperative Adverse Events to Improve Postoperative Care

**DOI:** 10.3390/jcm12072546

**Published:** 2023-03-28

**Authors:** Larsa Gawria, Ahmed Jaber, Richard Peter Gerardus Ten Broek, Gianmaria Bernasconi, Rachel Rosenthal, Harry Van Goor, Salome Dell-Kuster

**Affiliations:** 1Department of Surgery, Radboud University Medical Centre, 6525 GA Nijmegen, The Netherlands; 2Basel Institute for Clinical Epidemiology and Biostatistics, University Hospital Basel, University of Basel, 4051 Basel, Switzerland; 3Department of Surgery, Yitzhak Shamir Medical Centre, Tel Aviv 7030083, Israel; 4Clinic for Anesthesiology and Pain Therapy, Hospital of Fribourg, 1752 Fribourg, Switzerland; 5Faculty of Medicine, University of Basel, 4001 Basel, Switzerland; 6Clinic for Anesthesiology, Intermediate Care, Prehospital Emergency Medicine and Pain Therapy, University Hospital Basel, 4031 Basel, Switzerland; 7Department of Clinical Research, University of Basel, 4031 Basel, Switzerland

**Keywords:** intraoperative adverse events, intraoperative complications, origin of Intraoperative adverse events, classification of intraoperative adverse events

## Abstract

Background: Intraoperative adverse events (iAEs) are associated with adverse postoperative outcomes and cause a significant healthcare burden. However, a critical appraisal of iAEs is lacking. Considering the details of iAEs could benefit postoperative care. We comprehensively analyzed iAEs in a large series including all types of operations and their relation to postoperative complications. Methods: All patients enrolled in the multicenter ClassIntra^®^ validation study (NCT03009929) were included in this analysis. The surgical and anesthesia team prospectively recorded all iAEs. Two researchers, blinded to each other’s ratings, appraised all recorded iAEs according to their origin into four categories: surgery, anesthesia, organization, or other, including subcategories such as organ injury, arrhythmia, or instrument failure. They further descriptively analyzed subcategories of all iAEs. Postoperative complications were assessed using the Comprehensive Complication Index (CCI^®^), a weighted sum of all postoperative complications according to the Clavien–Dindo classification. The association of iAE origins in addition to the severity grade of ClassIntra^®^ on CCI^®^ was assessed with a multivariable mixed-effects generalized linear regression analysis. Results: Of 2520 included patients, 778 iAEs were recorded in 610 patients. The origin was surgical in 420 (54%), anesthesia in 283 (36%), organizational in 34 (4%), and other in 41 (5%) events. Bleeding (*n* = 217, 28%), hypotension (*n* = 118, 15%), and organ injury (*n* = 98, 13%) were the three most frequent subcategories in surgery and anesthesia, respectively. In the multivariable mixed-effect analysis, no significant association between the origin and CCI^®^ was observed. Conclusion: Analyzing the type and origin of an iAE offers individualized and contextualized information. This detailed descriptive information can be used for targeted surveillance of intra- and postoperative care, even though the overall predictive value for postoperative events was not improved by adding the origin in addition to the severity grade.

## 1. Introduction

Intraoperative adverse events (iAEs) are relevant to postoperative care and quality improvement. One-half to two-thirds of all perioperative events are attributed to surgical care, with the majority occurring during surgery and more than one-half of these appearing to be preventable [1,2,3]. Awareness for safe intraoperative care is raised with the emergence of minimally invasive surgery, the increased complexity of operations, and the higher number of elderly and multimorbid surgical patients [4,5,6]. Standardized reporting of iAEs is key for identification of repeated occurrence of events and for improving perioperative care [7]. Compared to the reporting of postoperative complications, for which the Clavien–Dindo classification is dominantly applied, iAEs lag behind in uniform and standardized reporting in clinical practice and the available literature [7,8]. This is reflected by the 270-times more cited Clavien–Dindo classification compared to all available classifications of iAEs together [9].

Generally, the operative and anesthesia report is used to report and describe iAEs. However, operative reports have been found to be subjective; events are underreported; and reports rarely include organizational causes such as equipment failure [10,11]. In addition, operative reports may be delayed, resulting in incomplete handovers at transfers to higher-level care or surgical wards [11,12]. Several grading systems for iAEs have been developed. These systems usually have several important drawbacks that hinder their uniform implementation (e.g., not including all sources of iAEs, focusing on specific operations (laparoscopic) or specific iAEs (adhesiolysis), and being complex or not properly validated) [13,14,15,16]. Our group recently developed and validated ClassIntra^®^, an easy-to-use grading system for all types of iAEs, and found a strong association between the severity of intraoperative and postoperative complications in a range of surgical disciplines [17]. This association was further established for visceral surgery [18]. Similar to other grading systems, ClassIntra^®^ does not describe the origin of the iAE. Such a description might add context to the severity grading and possibly strengthen the association with the postoperative outcome. The context can also improve postoperative handovers and early diagnosis of postoperative complications, and may serve as a tool for training and quality improvement [19,20].

The large prospective database of the ClassIntra^®^ study offers the possibility to describe the attributes of iAEs including origins and subcategories as a means for improved postoperative handovers and the development of strategies to prevent postoperative complications. We hypothesized that the addition of the origin to the severity grade of an iAE according to ClassIntra^®^ could strengthen the association between the severity grade of the iAE and the postoperative complication. Therefore, we evaluated the prognostic value of the origin of iAEs on postoperative complications when added to the ClassIntra^®^ grading system.

## 2. Materials and Methods

Operative data of an international study aimed at validating the ClassIntra^®^ classification for iAEs was used in this analysis [17]. Eighteen centers from 12 countries prospectively enrolled 2520 consecutive in-hospital patients undergoing any type of surgery in whom iAEs were reported and graded according to ClassIntra^®^ (Appendix A).

This classification defines an iAE as any deviation from the ideal intraoperative course that occurs between skin incision and skin closure, and consists of five severity grades depending on the required intervention and patient symptoms. The attending surgical and anesthesia teams reported the severity grade and a free-text description of the iAE(s) directly after surgery. Patients were assessed daily for postoperative complications until hospital discharge and had one post-discharge follow-up to assess 30-day mortality. Postoperative complications were assessed and graded according to the Clavien–Dindo classification by the physician on the ward [8,21]. A weighted sum of all postoperative complications in a single patient was calculated using the Comprehensive Complication Index (CCI^®^) [22,23]. The CCI^®^ forms a continuous scale from 0 (no complications) to 100 (postoperative death) based on grades according to Clavien–Dindo [23]. All participating centers in the validation study provided consent to use their data for this study. No approval was required from the local ethical committees of the study centers in addition to the existing approval for the ClassIntra^®^ study (EKNZ Req-2016-00469; ClinicalTrials.gov, NCT03009929).

### 2.1. Categorization

Free-text descriptions of the iAEs were evaluated to identify the origin of the iAEs. The origin was categorized into four categories: surgical, anesthesia, organizational, and other. Surgical iAEs were defined as events initially arising in the operative field, such as bleeding or an iatrogenic bowel injury. Anesthesia-related iAEs included all medical events not arising in the operative field (e.g., arrhythmia or hypoxemia). iAEs that involved more than one origin were categorized according to the origin causing the sequela. For example, hypotension caused by bleeding was categorized as surgical, while hypotension resulting from anaphylaxis was categorized as anesthesia.

Organizational iAEs were due to errors in logistic or technical failure (e.g., instrument failure). iAEs were categorized as ‘other’ in cases where the origin was not clear from the description, or if they occurred before skin incision or after skin closure and did not match any set definition [17].

The list of subcategories was designed as an open list with subcategories added to further describe the parent category (e.g., the type of bleeding or hypotension) when appropriate, as outlined below.

As bleeding and hypotension were common heterogeneous subcategories with a range of treatments, the respective iAEs were further specified. For bleeding, a differentiation was made between diffuse and major. If the description stated “minor”, “small”-vessel or “diffuse”, bleeding was classified as ‘diffuse’. When a large caliber vessel was indicated, either by naming the vessel or with the terms “large” or “major”, bleeding was classified as ‘major’. In case of hypotension, a differentiation was made between mild, profound, or unknown severity, based on the required treatment as mentioned in the description of the iAEs. If the description noted ephedrine or phenylephrine, or “mild” or “transient”, hypotension was classified as ‘mild’. If the description noted noradrenaline or “strong”, hypotension was classified as ‘profound’. A description that did not distinguish the type of bleeding or hypotension was left unspecified. The attending team reported conversion from minimally invasive to open surgery when they judged it as an iAE. We categorized this based on the provided context in the free text.

Categorization and subcategories were recorded by two researchers (LG and AJ) who were blinded for each other’s assessments. Thirty iAEs were used for training. An intraclass correlation coefficient was used as a reliability measure for categorization of the origin. In case of differences in origins or subcategories, two senior physicians (RtB and SDK) were consulted for surgical and anesthesia iAEs, respectively, to reach consensus. Categorization was recorded in a Microsoft Access database for Office 365, which included the iAEs and relevant patient-related information.

### 2.2. Outcomes and Statistical Analysis

We used descriptive statistics and frequency tables of all iAEs and their distribution across origins and subcategories.

In an explorative way, we investigated the effects of the origin of an iAE on the CCI^®^ in addition to the severity grade, using multivariable linear mixed-effect regression analyses [23]. The multivariable models with and without the origin of iAEs were compared using a likelihood ratio test. We tested the interaction between the origin and ClassIntra^®^ grade, which was not significant and therefore not included in the model. No extensive testing of the model’s predictive ability was conducted due to the exploratory nature.

As more than one iAE of different origins could occur in one patient, we categorized the origin variable for statistical analyses into the following 5 levels: 1 = no iAE, 2 = surgical origin of a single iAE, 3 = anesthesia origin of a single iAE, 4 = organizational origin of a single iAE, 5 = in case more than one iAE of any origin occurred. iAEs in the other origin category were not taken into account as these were insufficiently described or were not considered iAEs according to pre-set definitions [17].

The model was adjusted for predefined potential confounders: patient age, American Society of Anesthesiologists (ASA) physical status [24], complexity graded as one of five categories (minor, intermediate, major, major plus, and complex major operation) according to the British United Provident Association (BUPA) [25], the duration and urgency of the surgical procedure, the wound category [26], and the experience of the surgery and anesthesia teams. The variables for anesthesia and surgical experience were handled as in the validation study of ClassIntra^®^ [17]. In short, anesthesia experience was summed up with anesthesia nurse in training, his/her graduation, and a resident present in the operating room each contributing one point; a consultant added another 2 and a senior consultant added 3 points. Surgical experience was defined by the most senior surgeon present in the operating room, to which the consultant and the resident (in training) were compared.

Complexity grades were not available for 4% of the procedures, for which an alternative grade corresponding to a comparable procedure was used. There were no other missing data. All analyses were performed using Stata/SE 15.1 for Windows (StataCorp, College Station, TX, USA). We followed the STROBE guidelines for reporting the results.

## 3. Results

Out of 2520 patients, 610 (24%) experienced 778 iAEs according to ClassIntra^®^, of which 198 (25%) were of grade I, 417 (54%) grade II, 142 (18%) grade III, and 21 (2.7%) grade IV. No intraoperative deaths of grade V occurred. Baseline characteristics and postoperative outcomes are described in Table 1 and Table 2.

### 3.1. Origin of iAEs

Of all 778 iAEs, the researchers classified a total of 420 (54%) iAEs of surgical origin, 283 (36%) of anesthesia origin, 34 (4.4%) of organizational origin, and 41 (5.0%) of other origin (Figure 1).

Frequency of iAE by severity grade according to ClassIntra^®^ are depicted in Figure 1.

All grades, except for grade II, were more frequently reported with surgery as the origin as opposed to anesthesia, with grade I at 113 (27%) vs. 55 (19%), grade III at 84 (20%) vs. 52 (18%), and grade IV at 12 (3.1%) vs. 5 (1.8%), respectively. Grade II was less frequently observed to have a surgical origin as compared to anesthesia, with 210 (50%) vs. 172 (60%). Although iAEs with an organizational origin were predominantly of grade I and II, at 18 (53%) and 14 (41%), respectively, we note that 2 (5.8%) grade III iAEs occurred, meaning patients with severe symptoms that were potentially life-threatening. iAEs in the other category occurred before incision or after skin closure and were outside the definitions of ClassIntra^®^ (34, 87%); referred to open-close procedures due to unresectable tumors (3, 7.7%); or were insufficiently described (2, 5.1%). Out of the iAEs that were outside the window, 5/34 (15%) were severe, i.e., grade III or IV.

In the case of unplanned procedures, e.g., emergency or urgent, iAEs of surgical origin occurred twice as often compared to anesthesia, in 65 (15%) and 19 (6.7%) cases, respectively.

Gastrointestinal surgery was the largest discipline and included most iAEs compared to the other surgical disciplines in Table 2.

However, the distribution of the iAE origins varied per discipline.

Regarding the distribution of origin according to the case-mix of patients, surgery-related iAEs occurred more than anesthesia-related iAEs in ASA I patients, in 63/430 (15%) and 13/283 (4.2%) cases, respectively (see Table 1). However, for ASA IV patients the incidence rate was reversed. There were fewer surgical and more anesthesia iAEs, with 21/420 (5.0%) and 32/283 (11%) cases, respectively. Likewise, fewer unplanned ICU postoperative admissions were reported after surgical iAEs compared with anesthesia, with 170/420 (40%) and 172/283 (60%) cases, respectively, per Table 3.

The experience of the surgical or anesthesia teams did not differ among the iAE origins.

A total of 68 (8.7%) iAEs involved more than one origin. Of these, 35 (52%) involved hypotension due to bleeding following an inadvertent injury. In these cases, the origin of the discipline that caused the sequela of the iAEs was accounted for.

The intraclass correlation coefficient for the origins of iAEs between both researchers was 0.60 (95% CI 0.55–0.64). Full consensus was reached after expert consultation.

### 3.2. Subcategories of Origin of iAEs

Bleeding was the most frequent iAE of surgical origin, with 217 (28%) cases as shown in Table 4.

Approximately one-third of the specified bleeding iAEs were of a major caliber vessel, with 33 (28%) cases. Six bleeding iAEs were of grade IV which needed major and urgent treatment because of life-threatening symptoms, of which five were specified as major and one was unspecified. A similar frequency of major caliber bleeding was observed when categorized by emergency and elective operations.

Organ injury was the second most frequent subcategory of iAEs of surgical origin and included a quarter of the surgical iAEs mainly of low severity, with grade I at 25/98 (26%) and grade II at 57/98 (58%) cases. The majority of injuries were serosa lesion, enterotomy, and gallbladder injury. Adhesiolysis was mentioned in 33 events; in 13 of these cases, adhesiolysis coincided with organ injuries such as serosa injury or enterotomy. A total of 13 (3%) conversions were reported, of which 5 were due to limited overview, 1 to bleeding, 1 to instrument failure, and 6 with no provided context.

In anesthesia, cardiovascular iAEs were most often reported with 210 (67%) cases, including 118 (56%) cases of hypotension and 31 (15%) cases of arrhythmia. Based on the required treatment for hypotension, 53 (45%) cases were mild, 36 (31%) were profound, and 29 (25%) were unspecified. A total of 21 cases of hypotension were severe (grade III or IV) of which 20 were profound and 1 was unspecified. Mild hypotension iAEs were recorded with low severity, namely, 9 grade I and 44 grade II cases. In addition, the unspecified cases were mostly low severity with 5 grade I, 23 grade II, and 1 grade III case.

A total of 28 iAEs were organizational: 12 (43%) were due to instrument failure, 10 (36%) were due to team communication, and 6 (21%) were due to logistics, all of which were grade I or II.

### 3.3. Multivariable Analysis

The log-likelihood ratio test comparing goodness of fit of the multivariable models including severity grades of ClassIntra^®^ with and without origin was not statistically significant (*p* = 0.15; Appendix A).

## 4. Discussion

The descriptive analysis of 778 iAEs in 610 patients, from an international multicenter prospective cohort study across a wide range of surgical disciplines and anesthesia, offered insights in the incidence and origin of iAEs that occurred between skin incision and skin closure. Surgery encompassed half of all iAEs, and anesthesia accounted for one-third. Almost one in ten iAEs involved both disciplines and seemed interdependent. Organizational iAEs were rarely reported, likely due to the lack of awareness of the origin as an iAE, but still viewed as part of the procedure. Bleeding, hypotension, and organ injury were the most frequently reported subcategories of origin. The addition of origin did not alter the previously reported association between severity of iAEs and postoperative complications [17,18].

The detailed analyses and work-up of the origin of iAEs offer important advantages. While surgery accounted for the majority of iAEs, the proportion of the most severe iAEs was comparable with anesthesia-related iAEs. One in five of surgery- and anesthesia-related iAEs was of major severity, defined as ClassIntra^®^ grade III or IV, which potentially leads to permanent disability. The reporting of well-recognized iAEs (e.g., bleeding and hypotension) was close to reality as reflected by the high incidence. However, with the increasing complexity of procedures and the usage of minimally invasive surgical devices, more organizational device-related iAEs were expected but not reported. Only 12 out of 778 iAEs (1.5%) were reported as instrument failure, which is a fraction of the 15% incidence that was reported by direct observation using audio and video recorders, also known as medical data recording [27]. Organizational iAEs might be of lower severity but impact the duration of surgery [28]. Medical data recording of laparoscopic cholecystectomies revealed an average delay of 15 min for each procedure due to workflow interruptions, with a subsequent increase in financial health costs [28].

All study centers participating in the ClassIntra^®^ validation study routinely used a perioperative checklist and an enhanced recovery protocol after surgery whenever applicable for the type of surgery. Yet, a standardized system for reporting iAEs was not developed, possibly leading to small differences in the reported incidences of iAEs between centers. Surgeons have indicated that the most common barriers to reporting iAEs are the fear of litigation, the lack of a standardized reporting system, and the absence of clear definitions for iAEs [29]. Longstanding systemic and cultural practices have hampered adequate reporting of iAEs, but this could be overcome with a positive culture and open communication surrounding iAEs [9,30]. A validated grading system offers a tool for uniform and standardized reporting but falls short of addressing the details of the iAE that could be relevant for postoperative care. Grading an iAE including its origin and describing subcategories may offer structured, relevant, and complete information for postoperative debriefing and handover to the recovery room, general ward, or intensive care unit. More complete information may avoid communication failure at the postoperative handover, which is the root cause for 70% of sentinel events in the postoperative course [31]. The simplicity of the ClassIntra^®^ classification with origin and subcategories allows for easy integration in the sign-out of the WHO safety checklists directly after surgery [17,32].

There is a rapid increase in the number of publications investigating iAEs but comparisons are impeded by their heterogeneity [9]. More than 20 different definitions for iAEs are applied and methods vary from chart reviews, (prospective) self-reporting, direct observation by human observers, to medical data recording [3,4,7,17].

With the emergence of checklists, crew resource management protocols, and medical data recording in the operating room, non-surgical iAEs of anesthesia- and non-technical origin (e.g., organizational or communication) have also gained interest [17,33,34,35]. A prospective evaluation of the characteristics of iAEs of any origin in a large cohort of multiple surgical disciplines is new and may overcome shortcomings of previous studies.

For example, Kaafarani et al. conducted a retrospective chart review of surgery-related iAEs in abdominal surgery and developed a classification for iAEs [14]. They identified an iAE incidence of 1.9%, which is much lower than the 17% incidence of surgery-related iAEs in this study. Moreover, the study did not account for anesthesia-related iAEs and, hence, ignored a significant part of the intraoperative course.

A landmark study by Gawande et al. conducted two decades ago investigated 15,000 surgical patients for iAEs of surgical or other medical origins, including anesthesia. Although Gawande accounted for a range of iAEs, e.g., bleeding, dysrhythmia, acute myocardial infarction, and technique-related complications, they did not differentiate between intra- and postoperative events. Overall, they found that more than half of all iAEs were of surgical origin, of which half were deemed preventable [3]. Despite the lack of further details concerning iAEs, the reported incidences were considerably lower than the 24% incidence rate of iAEs reported in our study. This difference is possibly due to the increased awareness of the impact of iAEs on patient outcomes in the surgical and anesthesia community overall, which is also reflected by the broad implementation of perioperative quality improvement programs such as surgical safety checklists and enhanced recovery after surgery [32,36]. A study investigating reporting bias revealed twice as many intra- and postoperative complications by chart review, compared to self-reporting by the treating perioperative team [37]. The main strength of this study is the detailed information of any type of iAEs in a large and broad surgical cohort across countries, improving the generalization of results. The high incidences reported most likely reflect real occurrences due to the prospective nature of this study and the motivation of participating clinicians to record iAEs [38].

However, this study also has limitations. First, surgical and anesthesia teams may have had different behavior towards a certain event type with a higher interest and knowledge of surgical and anesthesia in contrast to organizational iAEs. This may question the accuracy of the reported incidence of the latter event type [37]. Second, categorization of iAEs may have been wrong in some cases due to the limited context provided in the free-text description, despite two blinded clinical researchers and consensus in all cases after consultation with senior physicians. In addition, categorization of hypotension might be flawed as the optimal blood pressure for adequate perfusion is not individually weighted. Third, an iAE may arise due to an interplay of causal factors including organizational, human, and patient-related factors [39]. In this study, 10% of all iAEs involved multiple origins. Our data did not allow for describing the interaction between the different causative factors of iAEs. In particular, discussing interdependent iAEs with all members of the operative team can reveal insights in the pathogenesis of iAEs and trigger concerted postoperative diagnostic and therapeutic measures, which may enable early decision making to prevent postoperative complications and longer hospital stays [12]. Finally, we acknowledge that adverse events could have occurred outside the defined window between skin incision and skin closure. The definition for this timeframe is based on the results of the Delphi process in which ClassIntra^®^ (formerly CLASSIC) was developed [40]. An additional study is planned to reevaluate the timeframe for assessing iAEs and to extend it beyond skin incision until skin closure.

Introducing content-rich, uniform and adequate reporting, and a positive learning culture allows for benchmarking of iAEs in clinical practice and research. It could enhance open communication and efforts for the development and implementation of strategies to mitigate iAEs.

## 5. Conclusions

Adding origin and subcategories to the severity grade of ClassIntra^®^ may offer individualized and contextualized information of iAEs, directing surveillance in the postoperative care, however, without altering the prognostic strength of this classification. Simple and complete descriptions of iAEs might be most relevant for easing the postoperative debriefing, handovers, and decision making on the ward or in the ICU.

## Figures and Tables

**Figure 1 jcm-12-02546-f001:**
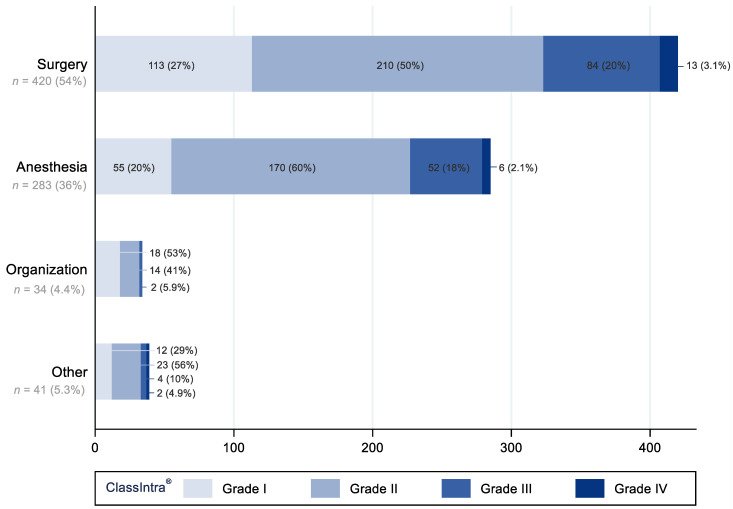
Origin of intraoperative adverse events versus severity grade of intraoperative adverse events according to ClassIntra^®^. Frequencies and percentages are displayed according to grade.

**Table 1 jcm-12-02546-t001:** Baseline characteristics for the total study population (*n* = 2520) and for subgroups without intraoperative adverse events (iAEs) (*n* = 1910) and with at least one iAE (*n* = 610).

	All Patients (*n* = 2520)	Patients without iAEs (*n* = 1910, 76)	Patients with iAE (*n* = 610, 24)
American Society of Anesthesiologists (ASA) physical status			
ASA I	503 (20)	431 (23)	72 (12)
ASA II	1118 (44)	852 (45)	266 (44)
ASA III	805 (32)	565 (30)	240 (39)
ASA IV	92 (4)	62 (3)	30 (5)
ASA V	2 (0.1)	-	2 (0.3)
Age in adults, median (IQR, range) (*n* = 2340)	61 (46–72; 18–97)	60 (45–71; 18–97)	64 (49–74; 18–93)
Sex			
Male	1382 (55)	1038 (54)	344 (56)
Female	1138 (45)	872 (46)	266 (44)
Body Mass Index in adults (kg/m^2^), median (IQR) (*n* = 2340)	26 (23–30)	26 (23–30)	26 (23–30)
Surgical discipline			
Gastrointestinal surgery	1437 (57)	1085 (57)	352 (58)
Orthopedic surgery and traumatology	297 (12)	260 (14)	37 (6)
Vascular surgery	169 (7)	121 (6)	48 (8)
Urology	134 (5)	109 (6)	25 (4)
ENT and maxillofacial surgery	122 (5)	99 (5)	23 (4)
Neuro- and spine surgery	96 (4)	53 (3)	43 (7)
Cardiac surgery	73 (3)	41 (2)	32 (5)
Pediatric surgery	54 (2)	48 (3)	6 (1)
Gynecology	46 (2)	29 (2)	17 (3)
Obstetrics	44 (2)	31 (2)	13 (2)
Reconstructive and hand surgery	26 (1)	21 (1)	5 (1)
Thoracic surgery	22 (1)	13 (1)	9 (2)
Complexity of surgical procedure			
Minor	105 (4)	94 (5)	11 (2)
Intermediate	437 (17)	383 (20)	54 (9)
Major	790 (31)	613 (32)	177 (29)
Major plus	442 (18)	323 (17)	119 (20)
Complex major operation	648 (26)	431 (23)	217 (36)
Urgency of procedure			
Planned	2153 (85)	1627 (85)	526 (86)
Unplanned	367 (15)	283 (15)	84 (14)
Operating surgeon			
Senior consultant	1662 (66)	1239 (65)	423 (69)
Junior consultant	544 (22)	427 (22)	117 (19)
Resident	314 (12)	244 (13)	70 (11)
Anesthesia consultant present	2311 (92)	1746 (91)	565 (93)
Senior consultant	1481/2311 (64)	1112/1746 (64)	369/565 (65)
Junior consultant	830/2311 (36)	634/1746 (36)	196/565 (35)

All values are frequencies and percentage (*n*, %) unless stated otherwise. ENT = ear, nose, throat surgery.

**Table 2 jcm-12-02546-t002:** Origin of intraoperative adverse events (iAEs) according to surgical discipline. Multiple iAEs are possible in one patient. All values are frequencies and row percentages. (*n*, %).

	Origin				
Disciplines	Total iAEs (*n* = 778, 24)	Surgery (*n* = 420, 54)	Anesthesia (*n* = 283, 36)	Organization (*n* = 34, 4.4)	Other (*n* = 41, 5.3)
Gastrointestinal surgery (*n* = 1437)	442 (24)	289 (65)	117(26)	17 (4)	19 (4)
Orthopedic surgery (*n* = 297)	40 (11)	18 (45)	19 (48)	1 (3)	2 (5)
Vascular surgery (*n* = 169)	64 (28)	35 (55)	24 (38)	2 (3)	3 (5)
Urology (*n* = 134)	29 (18)	3 (10)	17 (59)	2 (7)	7 (24)
Ear, nose, throat and maxillofacial surgery (*n* = 122)	25 (19)	9 (36)	12 (48)	2 (8)	2 (8)
Neuro- and spine surgery (*n* = 96)	58 (45)	15 (26)	38 (66)	2 (2)	3 (5)
Cardiac surgery (*n* = 73)	62 (44)	26 (42)	36 (58)	-	-
Pediatric surgery (*n* = 54)	6 (11)	5 (82)	1 (17)	-	-
Gynecology (*n* = 46)	22 (37)	7 (32)	8 (36)	7 (32)	-
Obstetrics (*n* = 44)	16 (30)	5 (32)	9 (56)	-	2 (13)
Reconstructive and hand surgery (*n* = 26)	5 (19)	2 (40)	2 (40)	1 (20)	-
Thoracic surgery (*n* = 22)	9 (41)	6 (67)	2 (22)	-	1 (11)

**Table 3 jcm-12-02546-t003:** Postoperative outcomes for the total study population (*n* = 2520) and for subgroups without iAEs (*n* = 1910) and with at least one iAE (*n* = 610).

	All Patients (*n* = 2520)	Patients without iAEs (*n* = 1910, 76)	Patients with iAE (*n* = 610, 24)
Origin of procedure (several iAEs per patient possible)			
No iAE	1910 (71)	1910 (100)	-
Surgery	420 (16)	-	420 (54)
Anesthesia	283 (11)	-	283 (3)
Organization	34 (1.3)	-	34 (4.4)
Other	41 (1.6)	-	41 (5.3)
Most severe iAE according to ClassIntra^®^			
0	1910 (76)	1910 (100)	-
Grade I	161 (6.4)	-	161 (6.4)
Grade II	309 (12)	-	309 (12)
Grade III	122 (4.8)	-	122 (4.8)
Grade IV	19 (0.8)	-	19 (0.8)
Grade V	-	-	-
Most severe postoperative complication			
0	1682 (67)	1367 (72)	315 (52)
Grade I	349 (14)	257 (13)	92 (15)
Grade II	277 (11)	162 (8.5)	115 (19)
Grade IIIa	72 (2.9)	45 (2.4)	27 (4.4)
Grade IIIb	55 (2.2)	40 (2.1)	15 (2.5)
Grade IVa	53 (2.1)	23 (1.2)	30 (4.9)
Grade IVb	7 (0.3)	3 (0.2)	4 (0.7)
Grade V	25 (1.0)	13 (0.7)	12 (2.0)
Duration of surgery, median (IQR, range)	100 (60–170, 4–760)	90 (55–147, 4–760)	151 (93–230, 12–673)
Postoperative length of hospital stay, median (IQR, range)	3 (2–6, 0–191)	3 (1–5, 0–106)	6 (3–9, 1–191)
IMC/ICU during postoperative course	68 (2.7)	40 (2.1)	28 (4.6)
Intermediate care unit (IMC)	18 (26)	15 (38)	3 (11)
Intensive care unit (ICU)	50 (74)	25 (63)	25 (89)
30-day mortality	26 (1.1)	13 (0.7)	13 (2.1)

All values are frequencies and percentage (*n*, %) unless stated otherwise. iAEs = intraoperative adverse events.

**Table 4 jcm-12-02546-t004:** Origin and subcategories of origin of intraoperative adverse events according to severity graded by ClassIntra^®^. All values are frequencies and column percentages (*n*, %). Organiz. = organization, oth. = other. * Extensive adhesiolysis without organ injury.

		ClassIntra^®^				
	Subcategories	Total (*n* = 778)	Grade I (*n* = 198, 25%)	Grade II (*n* = 417, 54%)	Grade III (*n* = 142, 19%)	Grade IV (*n* = 21, 3%)
Surgery	Bleeding	217 (55)	65 (59)	99 (50)	47 (60)	6 (55)
	Diffuse	87 (40)	47	37	3	-
	Major	33 (15)	2	12	14	5
	Unspecified	97 (45)	16	50	30	1
	Organ injury	98 (25)	25 (23)	57 (29)	13 (17)	3 (27)
	Seromuscular	28 (29)	1	26	1	-
	Enterotomy	14 (14)	1	8	4	1
	Gallbladder	12 (12)	7	4	1	-
	Urinary system	7 (7)	1	3	3	-
	Spleen	6 (6)	1	3	-	2
	Pulmonal	6 (6)	1	4	1	-
	Liver	4 (4)	1	3	-	-
	Appendix	3 (3)	1	1	1	-
	Nerve	3 (3)	2	-	1	-
	Bone	3 (3)	1	1	1	-
	Stomach	2 (2)	1	1	-	-
	Other organ	10 (10)	7	3	-	-
	Adhesiolysis *	20 (5)	4 (4)	15 (8)	-	1 (9)
	Conversion	13 (3)	2 (2)	2 (1)	9 (12)	-
	Failed insertion of prosthesis	11 (3)	5 (5)	5 (3)	1 (1)	-
	Vessel anastomosis leak	6 (2)	-	4 (2)	2 (2)	-
	Bowel anastomosis leak	4 (1)	-	1 (1)	2 (3)	1 (9)
	Other surgical ****	29 (7)	9 (8)	16 (8)	4 (5)	-
Anesthesia	Cardiovascular circulation	210 (67)	42 (71)	122 (68)	41 (68)	5 (63)
	Hypotension	118 (56)	21	76	20	1
	Hypertension	26 (12)	8	14	4	-
	Arrhythmia	31 (15)	10	11	8	2
	Heart insufficiency	19 (9)	-	12	6	1
	Bradycardia	8 (4)	1	6	1	-
	Tachycardia	5 (2)	2	3	-	-
	Other cardiovascular	3 (1)	-	-	2	1
	Airway and respiratory system	28 (9)	4 (7)	13 (8)	9 (15)	1 (13)
	Hypoventilation	11 (39)	2	4	5	-
	Intubation related	7 (25)	-	5	2	-
	Hypoxemia	3 (11)	-	3	-	-
	Other airway related	7 (25)	1	1	2	1
	Laboratory findings	25 (8)	3 (5)	14 (8)	7 (12)	1 (13)
	Insufficient sedation	14 (5)	2 (3)	12 (7)	-	-
	Conversion to general anesthesia	5 (31)	-	5	-	-
	Need for extra sedation	9 (56)	2	7	-	-
	Systemic reactions	9 (3)	3 (5)	6 (3)	-	-
	Hypothermia	4 (44)	1	3	-	-
	Anaphylaxis	3 (33)	1	2	-	-
	Hyperthermia	2 (22)	1	1	-	-
	Renal system	4 (1)	1 (2)	3 (2)	-	-
	Lesions	2 (1)	1 (2)	1 (1)	-	-
	Pressure marks	1 (50)	-	1	-	-
	Other lesions	1 (50)	1	-	-	-
	Other anesthesia ****	19 (6)	3 (5)	12 (6)	3 (5)	1 (13)
Organiz.	Instrument failure	12 (43)	7 (41)	5 (45)	-	-
	Team communication	10 (36)	8 (47)	2 (18)	-	-
	Logistics	6 (21)	2 (12)	4 (36)	-	-
Oth.	Other	41 (5)	12 (6)	23 (6)	4 (3)	2 (10)

**** See Appendix A for descriptions of other surgical and anesthesia iAEs (Appendix A).

## Data Availability

Data is available upon special request. Anonymized patient level data are available for investigators whose proposed use of the data has been approved by a review committee identified for this purpose.

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
