# Peer review of "Appraisal of Intraoperative Adverse Events to Improve Postoperative Care"

_jcm, 2023, doi:10.3390/jcm12072546_

Round 1

Reviewer 1 Report

Thank you for giving me the opportunity to review this well-designed and well-written paper. In the context of standardization of processes and quality improvement in perioperative care, IAEs is a topic of high interest. The prospective multicentric study brings valuable data extracted from a significant cohort of patients having various types of surgery. The IAEs categorization according to its origin may positively influence the shaping and remodeling of perioperative care processes. Clear and accurate methodology and scientific language. Figures and tables are appropriate. Pertinent conclusions. Recent and relevant references. No further recommendations.

Author Response

We thank the reviewer for the appreciation of our manuscript. We agree that iAEs are a topic of interest. Future studies are planned to implement ClassIntra® in daily clinical practice.

Reviewer 2 Report

I congratulated the authors for submitting a good study titled”

Appraisal of intraoperative adverse events to improve postoperative care” to exam the correlation between intraoperative adverse events and postoperative outcomes. The authors did very good job on study design, data collecting, results presentation and discussion. I have some comments:

1.     There is a flaw in the definition of intraoperative courses. P3L87 “…ideal intraoperative 86 course that occurs between skin incision and skin closure…” Some of the iAEs will be overlooked if the above time frame is defined as intraoperative course.  The intraoperative time should include the time when patient enters the operating room to patient leaves the operating room. Many iAEs could potentially happen during periods of anesthesia induction, surgical prepping, emerging from anesthesia after the incision is closed, and extubation. There are all in the category of iAE from anesthesia.

2.     The authors should have disclosed the anesthesia types such as reginal anesthesia, monitored anesthesia care (MAC) or general anesthesia (GA). Most people would agree that changing anesthesia plan i.e., conversion from regional or MAC to GA  is considered significant event  

3.     How is hypotension defined in the study? Do you use a specific threshold for SBP, MAP  or percentage change from baseline? What about hypertension? Is this should be considered as AE? Take one step further, do you have a minimal duration for  hypotension?

4.     The grading for bleeding is also too vague? The amount of blood loss is not necessarily related to the size of the blood vessels injured. Large good loss could resulted from oozing from a big surgical surface rather than from injury to a larger vessel.

5.     The category of iAE from surgery is not very well defined. The authors mentioned that large proportion of the gastrointestinal surgery were performed under laporoscopy(minimally invasive). Will conversion to open be considered a iAE?

6.     In cardiac surgery, did you include reopening  the chest in the OR considering in the iAEs?

Author Response

  1. There is a flaw in the definition of intraoperative courses. P3L87 “…ideal intraoperative 86 course that occurs between skin incision and skin closure…” Some of the iAEs will be overlooked if the above time frame is defined as intraoperative course.  The intraoperative time should include the time when patient enters the operating room to patient leaves the operating room. Many iAEs could potentially happen during periods of anesthesia induction, surgical prepping, emerging from anesthesia after the incision is closed, and extubation. There are all in the category of iAE from anesthesia.

Reply:

We acknowledge the point highlighted by the reviewer, that periods outside the time frame between skin incision and closure carry risks for adverse events. However, this definition of the timeframe as only between skin incision and skin closure is based on the results of the Delphi process in which ClassIntra® (former CLASSIC) was developed [1]. As we agree with the reviewer that relevant iAEs are also likely to happen during the anesthesia period, we are currently planning a study to reevaluate the timeframe for assessing iAEs. In this planned project, we will compare the prognostic ability of any intraoperative adverse events as assessed during the current timeframe (between skin incision and skin closure) with the prognostic ability of iAEs assessed during an extended timeframe (between start anesthesia until handover in the postanesthesia care unit) for predicting postoperative events.

For the current study, we made use of the prospective database of the ClassIntra® validation study, including valuable information of the intraoperative course between skin incision and closure based on the current definition and timeframe.

We made revisions in the discussion section to highlight this point as follow:

Page 12, line 358-362: “Finally, we acknowledge adverse events could occur outside the defined window between skin incision and skin closure. The definition for this timeframe is based on the results of the Delphi process in which ClassIntra® (formerly CLASSIC) was developed [40]. An additional study is planned to reevaluate the timeframe for assessing iAEs and to extend it beyond skin incision until skin closure.”

  1. The authors should have disclosed the anesthesia types such as reginal anesthesia, monitored anesthesia care (MAC) or general anesthesia (GA). Most people would agree that changing anesthesia plan i.e., conversion from regional or MAC to GA  is considered significant event  

Reply:

We agree with the reviewer and can confirm that a change of anesthesia plan was considered an iAEs by the attending anesthesiologist. This was subcategorized as insufficient sedation (table 4). Out of 16 iAEs that described insufficient sedation, 5 conversions were reported. In addition, 9 iAEs described the need of extra sedation, and 2 rapid decurarization. As rapid decurarization is likely to occur at the end of anesthesia and after skin closure, these two events are subcategorized under the origin 'Others'.

For eighty percent of the patients the used anesthesia technique was general anesthesia. Further information regarding the anesthesia type is disclosed in the original validation study of ClassIntra® [2].

To offer more insight in the subcategory of insufficient sedation, we adjusted table 4 on page 7 and 8, and figure 1 accordingly.

  1. How is hypotension defined in the study? Do you use a specific threshold for SBP, MAP  or percentage change from baseline? What about hypertension? Is this should be considered as AE? Take one step further, do you have a minimal duration for  hypotension?

Reply:

We agree with the important point that is mentioned by the reviewer and, hence, disclosed this as a limitation in the discussion section, page 12 line 349-351. The available literature is inconclusive regarding a specific threshold for hypotension, but tends towards a patient centered approach [3,4]. The generic definition for iAEs according to ClassIntra® is suitable for this approach. However, we have only as much details as provided by the grading (which certainly is based on extent and duration of the episode) and as freetext by the treating team. We agree that also hypertensive periods should be considered as an iAE. They are, therefore, listed in table 4 with 26/210 (12%).

  1. The grading for bleeding is also too vague? The amount of blood loss is not necessarily related to the size of the blood vessels injured. Large good loss could resulted from oozing from a big surgical surface rather than from injury to a larger vessel.

Reply:

We agree that the amount of blood loss is not always related to the size of the injured blood vessel. However, the severity grading of blood loss was done by the operating team consisting of the attending surgeon and anesthesiologist and based on patient symptoms and required intervention, as defined by the grading system of ClassIntra®. The definition for the grades did not set any limitation regarding the injured blood vessel and amount of blood loss. For subcategorisation of reported bleeding, we could only use the freetext information.

To avoid any unclarity we replaced “severity of bleeding” to “type of bleeding” in the methods and materials sections, page 4, line 116: The list of subcategories was designed as an open list with subcategories added to further describe the parent category (e.g. type of bleeding or hypotension).”

And page 4, line 128:A description that did not distinguish the type of bleeding or hypotension was left unspecified”

  1. The category of iAE from surgery is not very well defined. The authors mentioned that large proportion of the gastrointestinal surgery were performed under laporoscopy(minimally invasive). Will conversion to open be considered a iAE?

Reply:

The reason for conversion from minimal invasive to open surgery could be strategic or reactive. The attending team judged whether this was an iAE, and graded it as such. If sufficient information was provided describing the context and reason for the conversion, categorization was done accordingly. For example when instrument failure caused the sequel to conversion, this was categorized as an iAE of organizational origin. When no additional information was provided, we categorized this as an iAE of surgical origin. A total of 13 conversions were reported, of which 5 due to limited overview, 1 to bleeding, 1 to instrument failure, and 6 with no provided context.

For clarity, we replaced the explanatory heading of table 4 and included the following sentence in the materials and method section under paragraph categorization.

Page 4, line 128-130: “The attending team reported conversion from minimal invasive to open surgery when they judged this as an iAE. We categorized this based on the provided context in free text.”

In the result section we added the above-mentioned information, page 10, line 251-253: “A total of 13 (3%) conversions were reported, of which 5 due to limited overview, 1 to bleeding, 1 to instrument failure, and 6 with no provided context.”

  1. In cardiac surgery, did you include reopening  the chest in the OR considering in the iAEs?

Reply:

There was no reopening of the chest in the OR in cardiac surgery reported as iAE in this cohort. However, in case reopening the chest is required while the patient is in the operating room and under anesthesia, this could be reported as an iAE and graded according to the patient symptoms (probably grade 4). However, if the patient left the operating room and an adverse event occurs with the need to reopen the chest, this should be reported as a postoperative complication grade IIIb according to Clavien-Dindo [5,6].

Reviewer 3 Report

Dear Authors,

   Thank you for the time you invested in writing thIs manuscript. By further analyzing the prospective database on which ClassIntra grading system was developed and validated, the authors sought to determine on one hand the origin of intraoperative adverese events and on the other hand, whether or not their origin strengthened their association with the patient's postoperative course.

   And, albeit no further association was proven, the findings of this study are exceptionally significant in the modern era of patient safety and promoting less complications for patients undergoing surgery. The authors have performed a very difficult task, that of classifying intraoperative compliactions according to their origin, and provide as with a taxonomy that should be used as reference in future studies. Furthermore, the manuscript provided, has many similarities with the now proven concept of utilizing avaition safety in medicine, meaning that complications tend to be multifactorial and thus, all "parts" of surgery (e.g. surgeon, anesthesia, environment) should be thoroughly checked before, during and after the procedure.

   All in all, this manuscript may be "SHOULD READ" material, for anyone advocating for patient's safety, and even more, for patients undergoing surgery.

Author Response

We thank the reviewer for the valuable recognition of our work, providing insight in the occurrence of iAEs of any origin including subcategorization, either surgery, anesthesia, or organizational. We recommend the operative team to discuss all iAEs at the end of surgery, possibly included in the “sign out” according to the Surgical Safety Checklist [7]. A graded iAEs that reports origin and subcategory could increase vigilance in the monitoring of patients at risk to potentially diagnose and prevent imminent complications earlier.

Round 2

Reviewer 2 Report

Thank you for addressing my comments and concerns.